# The impact of soil transmitted helminth on malaria clinical presentation and treatment outcome: A case control study among children in Bagamoyo district, coastal region of Tanzania

Nahya Salim Masoud[1,2,3,4]*, Stefanie Knopp[3,4], Nicole Lenz[5], Omar Lweno[1], Ummi Abdul Kibondo[1], Ali Mohamed[1], Tobias Schindler[1,4,6], Julian Rothen[1,4,6], John Masimba[1], Alisa S. Mohammed[1], Fabrice Althaus[3,7], Salim Abdulla[1], Marcel Tanner[3,4], Claudia Daubenberger[4,6], Blaise Genton[8]

**1** Bagamoyo Research and Training Centre, Ifakara Health Institute, Bagamoyo, United Republic of Tanzania, **2** Department of Pediatrics and Child Health, Muhimbili University Health and Allied Sciences (MUHAS), Dar es Salaam, United Republic of Tanzania, **3** Department of Epidemiology and Public Health, Swiss Tropical and Public Health Institute, Basel, Switzerland, **4** University of Basel, Basel, Switzerland, **5** Food Microbial Systems, Risk Assessment and Mitigation Group, Agroscope, Bern, Switzerland, **6** Department of Medical Parasitology and Infection Biology, Swiss Tropical and Public Health Institute, Basel, Switzerland, **7** Health Unit, International Committee of the Red Cross (ICRC), Geneva, Switzerland, **8** Department of Ambulatory Care and Community Medicine, University of Lausanne, Lausanne, Switzerland

* nsalim@ihi.or.tz

## Abstract

### Background

Parasitic infectious agents rarely occur in isolation. Epidemiological evidence is mostly lacking, and little is known on how the two common parasites *Plasmodium* and soil transmitted helminths (STH) interact. There are contradictory findings in different studies. Synergism, antagonism and neutral effect have been documented between *Plasmodium* and STH. This study investigated the impact of STH on clinical malaria presentation and treatment outcome.

### Methods

A matched case control study with a semi longitudinal follow up according to World Health Organization (WHO) antimalarial surveillance guideline was done among children aged 2 months to 9 years inclusively living in western rural areas of Bagamoyo, coastal region of Tanzania. Cases were children with uncomplicated and severe malaria enrolled from the health facilities while controls were children with asymptomatic *Plasmodium* parasitemia enrolled from the same community.

**Data Availability Statement:** All relevant data are within the paper and its Supporting Information files.

**Funding:** This study received financial support from the European Commission (Contract/Grant agreement number: 241642) (http://ec.europa.eu/research/health/infectious-diseases/neglected-diseases/projects/014_en.html) in the frame of the IDEA project "Dissecting the Immunological Interplay between Poverty Related Diseases and Helminth infections: An African-European Research Initiative." The funders had no role in study design, data collection and analysis, decision to publish, or preparation of the manuscript.

**Competing interests:** The authors have declared that no competing interests exist.

## Results

In simple conditional regression analysis there was a tendency for a protective effect of STH on the development of clinical malaria [OR = 0.6, 95% CI of 0.3–1.3] which was more marked for *Enterobius vermicularis* species [OR = 0.2, 95% CI of 0.0–0.9]. On the contrary, hookworm species tended to be associated with increased risk of clinical malaria [OR = 3.0, 95% CI of 0.9–9.5]. In multiple conditional regression analysis, the overall protective effect was lower for all helminth infection [OR = 0.8, 95% CI of 0.3–1.9] but remained significantly protective for *E. vermicularis* species [OR = 0.1, 95% CI of 0.0–1.0] and borderline significant for hookworm species [OR = 3.6, 95% CI of 0.9–14.3]. Using ordinal logistic regression which better reflects the progression of asymptomatic *Plasmodium* parasitemia to severe malaria, there was a 50% significant protective effect with overall helminths [OR = 0.5, 95% CI of 0.3–0.9]. On the contrary, hookworm species was highly predictive of uncomplicated and severe malaria [OR = 7.8, 95% (CI of 1.8–33.9) and 49.7 (95% CI of 1.9–1298.9) respectively]. Generally, children infected with STH had higher geometric mean time to first clearance of parasitemia.

## Conclusion

The findings of a protective effect of *E. vermicularis* and an enhancing effect of hookworms may explain the contradictory results found in the literature about impact of helminths on clinical malaria. More insight should be gained on possible mechanisms for these opposite effects. These results should not deter at this stage deworming programs but rather foster implementation of integrated control program for these two common parasites.

## Author summary

Contradictory results have been published on the impact of intestinal worm infections on malaria and there is an ongoing debate on whether general anti-helminthic treatment is desirable. For the first time, we provide highlights for the case of *Plasmodium* and *Enterobius vermicularis* co-infection in Tanzanian children. In our findings, helminth infection was shown to be associated with a protective effect on the development of clinical malaria. However, when looking at the specific effect of different helminth species, *E. vermicularis* was indeed protective against clinical malaria but co-infection with hookworm was associated with uncomplicated and even more so severe malaria. We speculate that specific effects of STH species tend to be obscured when analysis is carried out with all helminths aggregated. The opposite effects on malaria of all helminths reported in previous studies could be due to the individual effect of the most prevalent helminth species within the studied population. Further studies are needed to understand the impact of helminth species in low and high intensity areas in different age- and at-risk groups to better advise helminth and malaria control programs.

## Background

Parasitic infections such as *Plasmodium* and soil transmitted helminths (STH) are highly prevalent in the tropical regions and their interaction has been debated in the scientific world [1–6].

The general prevalence of STH in Tanzania is estimated between 57–85 [7]. Local reports suggests that all regions have some level of infection which may go up to 100% in certain ecological settings [8]. Generally, high prevalence of STH has been documented in the island of Pemba, Zanzibar and parts of Northeastern Tanzania along the coast and lake zones [9]. There has been a dramatic decrease in malaria specific mortality following the scaling up of malaria control strategies despite the existing burden [10,11]. Malaria control scale-up has progressed on both mainland Tanzania and Zanzibar through the National malaria control program. The overall burden of parasitic infection has dropped in most of the endemic countries [12] indicating a substantial public health gain but hotspots still exist. The practical implication of this co-infection is potentially huge, especially in children [13,14]. Children usually suffer from higher prevalence and heavy parasitic loads of both *Plasmodium* and STH [15,16]. Deworming programs have been advocated to prevent helminth related morbidity [17]. The question of how the two parasitic infections interact and how deworming should continue is difficult to answer due to conflicting research findings. The co-infecting parasites can interact through different mechanism including resource competition, direct interference and immune mediated response [18]. Immune response induced by chronic helminth infection may modify immune response to a *Plasmodium* and alter infection and disease risk and vice versa [1,3,4,18,19]. Synergism as well as antagonism have been documented between *Plasmodium* and different species of STH [4,5].

Reviews conducted on epidemiology and interaction between helminths and malaria in humans [4,5] and validity of previous studies on association between helminths and the incidence of malaria have been critically discussed by Fernández et al [20]. In general, it has been shown that helminth infection increases the risk of *Plasmodium* infection and protects against severe manifestations. Both reviews suggested that hookworms are associated with malaria morbidity while *Ascaris lumbricoides* shows a protective effect [4,5]. Most of the co-infection studies have been conducted among older children and utilized Kato–Katz technique to isolate helminths eggs. A cross sectional study conducted at Alaba Kulito health centre in South Ethiopia found that STH had very little contribution to malaria severity and had no significant impact on clearance rate of *Plasmodium* infection [21]. A case control study among older children and adults aged 5–60 years in Tierralta, Colombia found a positive association of malaria and hookworm and a protective effect with *A. lumbricoides* [22]. In Kabale district, South West Uganda, an area of low malaria transmission, results suggest no evidence of association between helminths and risk of malaria [23]. A recent published cohort study conducted among the Entebbe Mother and Baby study (EMaBS) along the Northern shore of lake Victoria where malaria is endemic showed an increased burden of childhood malaria morbidy associated with hookworm helminth infection during pregnancy [24]. In Northwestern Tanzania, co-infection of hookworms and *P. falciparum* have been reported in school-aged children [25]. In the island of Zanzibar, United Republic of Tanzania, early helminth infection has been documented to negatively associate with malaria among the 6–23 month old children [26].

Several factors such as population age, geographical location, type and intensity of helminth and of *Plasmodium* infection, immune status and malaria clinical state may interplay to cause effect variations. Recent findings showed a wide spatial heterogeneity in the prevalence of malaria and STH co-endemicity within the regions and within countries in Sub Saharan Africa (SSA) emphasizing the need to implement integrated control programmes [27]. Most of the published studies were conducted in areas where the prevalence of STH is generally high (above 50%) and not much has been investigated on *Strongyloides stercoralis* and *Enterobius vermicularis* species. To our knowledge, this is the first case-control study designed to investigate the impact of STH on malaria clinical presentation, response to treatment and outcome and first report on the effect of *E. vermicularis* and *S. stercoralis* on malaria burden. It is also the first time that the whole childhood range (2months to 9 years) is covered. The results will

guide improved implementation of existing programs and stimulate a tailored approach for intergrated control programs, specifically in Tanzania.

## Materials and methods

### Ethics statement

The study was conducted under the IDEA study protocol which was approved by the institutional review boards of the Swiss Tropical and Public Health Institute (Swiss TPH; Basel, Switzerland) and the Ifakara Health Institute (IHI; Dar es Salaam, United Republic of Tanzania). The ethical approval for the conduct of the study was granted by the Ethikkomission beider Basel (EKBB; Basel, Switzerland; reference number 257/08) and the National Institute for Medical Research of Tanzania (NIMR; Dar es Salaam, United Republic of Tanzania; reference number: NIMR/HQ/R.8a/Vol. IX/1098).

Reporting of the study follows STROBE checklist (Strengthening the Reporting of Observational studies in Epidemiology) [28].

Sensitization meetings were conducted with the local district, community, schoolteachers, and health authorities to inform about the purpose, procedures, risk and benefits associated with the study. Study related procedures were implemented once the appropriate and adequate informed consent process had taken place. Informed consent form were signed off by the parents or his/her legally authorized representative of children aged ten years or younger (ten years being the age-limit of children eligible for inclusion in our study) [29–31]. Illiterate parents/ guardians were asked to bring witness who participated within the discussion prior to obtaining their thumbprints and witness signature.

Participants infected with helminth and/or malaria, or other medical conditions received appropriate treatment/referral according to the national treatment guidelines of Tanzania. Children with STH were treated with albendazole (400mg single oral dose) and those with asymptomatic *Plasmodium* parasitemia and uncomplicated malaria received artemether lumefantrine (ALU). Children with severe malaria received quinine injections until clinically stable and able to take ALU before seven days of completing quinine treatment as per stipulated malaria guideline at the time of the study. In order to investigate the impact of STH on clinical presentation and treatment outcome, children diagnosed with STH received a delayed antihelminth treatment at the end of study follow up (day 42). To prevent unnecessary complication, children were closely followed up for safety and those with heavy helminth load and severe disease received treatment prior to day 42. To ensure protocol adherence, parents were explained the purpose of the study and advised not to give antihelminth treatment to their children during the course of follow up. Parents were advised to bring their children to the nearest dispensary in case of clinical emergency or call the study clinician in case they have further questions on the study procedures. During the whole study period, clinical and field team were placed within the study area for field visits follow ups and safety. An emergency number and ambulance support were provided throughout the study to ensure safety of participants.

### Study area

The study was conducted between July 2011 and November 2012 in the western rural area of Bagamoyo district, coastal region of Tanzania about 20 to 60 kilometres from Bagamoyo town as described previously [31,32]. The study area is covered by a clinical surveillance system (CSS), part of the Ifakara Health Institute (IHI), Bagamoyo Research and Training Centre (BRTC) which works in close collaboration with the Bagamoyo District Hospital (BDH) officials to ensure quality health care delivery using its research platforms [33]. Four main villages were purposely selected considering the transmission of malaria and helminth. The villages

included were Kiwangwa, Msata, Mkange and Magomeni. In total, there were 5 health centres and 59 dispensaries within the district where all cases were assessed and treated. Main health centres for recruitment and follow up were Kiwangwa, Mkange and Msata. Severe cases were referred to the district referral hospital, BDH [34], situated at urban Bagamoyo city, Dunda ward which is close to Magomeni. Generally, the prevalence of malaria around Bagamoyo town is low as compared to the rural areas in the west [35]. Prior to the start of the study in June 2011, a deworming campaign took place among under five years and school aged children. Two mass long lasting insecticidal nets (LLIN) "catch-up" campaigns were implemented in Tanzania between 2009 and 2011. The first "catch-up" campaign was launched in 2009 and 8.7 million LLIN were distributed to children under five years [36,37]. The 2010 universal coverage campaign targeted all sleeping spaces not protected through the previous catch-up campaign and the Tanzania National Voucher Scheme (TNVS) [38].

## Study design

The study was part of the IDEA project, an African-European Research initiative, funded by European community, with the aim of dissecting the immunological interplay between poverty related diseases (malaria, tuberculosis (TB) and Human Immunodeficiency Virus (HIV)) and helminth infections [39]. A case-control component with a semi longitudinal follow-up was developed within the malaria arm. Cases were recruited in two different groups, namely children with severe malaria, and cases with uncomplicated malaria. Controls were asymptomatic children with *Plasmodium* parasitemia. The outcome was clinical presentation of malaria and presence or absence of a co-infection with at least one of the helminth species investigated. The longitudinal short term observational part of the study consisted of the assessment of response to anti-malaria treatment in the three groups according to World Health Organization (WHO) procedure (day 0, 1, 2, 3, 7, 14, 28, 42) [40].

## Participant recruitment and follow up visits

Inclusion criteria were age $\geq$ 2months to < 10 years, written informed consent as obtained from the parents/legally accepted representative, resident and willing to stay within the study area for at least two months of the study follow up. A case of severe malaria was defined according to WHO malaria case definitions. A case of uncomplicated malaria was defined as temperature $\geq$37.5 associated with positive blood slide for *Plasmodium* or positive malaria rapid diagnostic test (mRDT) and no criteria for severe malaria. Asymptomatic *Plasmodium* parasitemia was defined as no symptoms of malaria from day 0 to 7 (assessed by history and physical examination at day 0 and 7) and associated with positive blood slide for *Plasmodium* (or positive mRDT) at day 0 and 7.

A community cross sectional study was planned to recruit 100 asymptomatic *Plasmodium* parasitemia children (controls), aged 2 months to 9 years inclusively. Since the prevalence of *Plasmodium* parasitemia was known to be around 10% within the study area, 1,000 children from the community were to be screened as previously explained [31,32]. Considering the purposeful selection of the villages based on the burden of both parasitic infection, proportionate sampling was not done. Village heath care workers were inviting the parents/caregivers to the recruitment facilities for screening post community engagement and sensitization. Included children were assessed at each visit (recruitment and follow up visits) by a qualified, trained study clinician for signs and symptoms of malaria and other common diseases using a structured questionnaire designed for the study. Recruitment of uncomplicated malaria cases were conducted in the dispensaries, west side of the study area. Suspected severe malaria cases recruited from the Bagamoyo district hospital (BDH) received additional investigation in the

ward to exclude other medical conditions mimicking malaria syndrome such as respiratory, gastrointestinal, and other illdefined illnesses. All clinical follow up visits were conducted at the nearest health centres/dispensaries where children were recruited.

## Sample collection, diagnosis of *Plasmodium* and helminth infection

Sample collection and diagnostic procedures have been explained previously [29–31]. Briefly stool, urine and blood samples were collected and examined using a broad set of quality controlled diagnostic methods for common STH (*Ascaris lumbricoides*, hookworm, *Strongyloides stercoralis*, *Enterobius vermicularis*, and *Trichuris trichiura*), schistosoma species and *Wuchereria bancrofti*. A finger-prick blood sample (~1.0 ml) was collected and stored on ice immediately after collection until examination in the laboratory at IHI-BRTC. Blood slides and malaria rapid tests (mRDTs) were used for *Plasmodium* diagnosis. Thin and thick blood films were performed, air dried and Giemsa stained for detection and quantification of malaria parasites according to the IHI research laboratory standard operation procedures (SOP) as adopted from WHO [41]. The use of mRDTs supported screening and management of children at the field while waiting for blood films results. Polymerase chain reaction (PCR) technique was not performed on the positive follow up slides, hence unable to distinguish weather it was a recrudescence or relapse of malaria infection.

## Data management and statistical analysis

Clinical and laboratory data were double entered using the DMSys software (FDA approved for ICH/GCP clinical trials). The helminth species specific results derived by each method were entered into an electronic data base using Microsoft ACCESS 2010. The two datasets were transferred into STATA that was used for data analysis (version 13.0 software, Stata Corp LP; College Station, Texas, USA). Data management, conversion and classification for the helminth species and *Plasmodium* parasitemia were done as explained in previously published papers following WHO criteria [29,31].

Baseline data (demographic and helminth distribution) were summarized according to malaria clinical status. In the report investigating distribution and risk factors for *Plasmodium* and helminth co-infections [31], age, location (village) and education level (not schooling) were among the significant predictors for co-infection. These three variables were thus used for matching the cases and controls. Age was categorized as children less than 3 years, pre-school children aged 3–5 years and school-aged children from 6–9 years inclusively. The categorization was chosen to explore the age dependency variability considering the ongoing malaria and mass drug administration helminth programs with different approaches based on age, mainly focusing on children under five and above five years of age.

A matching program was developed with support from the STATA conference and users' group (http://www.stata.com/meeting/). The program reduces to the following syntax, whose arguments are explained below: radmatch varlist [id() mpair() case() k() rad() seed(). Briefly the program works as follows: i) first it creates a dataset of controls and cases separately from the variable that defines them (varlist) ii) then it randomly orders the cases using the unique identification id (varname) iii) a variable is then defined that linked a given control to its matched case, mpair() iv) using a loop over the observations in case and control datasets, matches are established wherever the cases and controls shares the same values, hence ending up with a dataset with all possible matches called (match) v) for each pair labelled 0 for a control and 1 for a case) vi) k() defines the k in the 1: k matching, the default is k = = 1 vii) rad() these are the respective radius corresponding to the variables specified in the varlist, viii) seed () this is the seed to replicate results.

Simple and multiple conditional logistic regression models were used to investigate the strength of association between helminth infection and malaria clinical status (from asymptomatic *Plasmodium* parasitemia to severe malaria). Odds ratios (OR) including 95% confidence interval and p-values were calculated. Most of other explanatory variables (gender, nutritional status and interventions used) were not significant in the simple regression analysis but were still included in the multiple conditional regressions as all of them are somehow expected to have an effect on the outcome. No interaction effect was observed with the matched variables used. The relationship between STH infection species and malaria clinical status was further explored using ordinal logistic regression model to better reflect the risk with increasing severity of malaria clinical status.

Treatment response was analyzed using the definitions of adequate clinical and parasitological response as advised by WHO [41]. Geometric mean time (in hours) to first parasite clearance according to helminth infection was estimated using a regression model adjusted for age group. The parasite counts were converted into log scale for analysis. The occurrence of diseases other than malaria was then summarized according to the helminth infection status. We could not analyze the correlation between STH species intensity and *Plasmodium* parasite density as most of the children presented with light intensity STH infection and insufficient numbers within the malaria infection clinical status.

## Results

Among 1,130 children whose parents gave consent and enrolled, 94 had asymptomatic *Plasmodium* parasitemia, 124 had uncomplicated malaria and 19 had severe malaria (Fig 1). The prevalence of helminth infection among children according to malaria clinical status before matching is described in Table 1. There was a reduced prevalence of helminth with increasing clinical malaria severity. Hookworm single helminth species co-infection significantly increased with increasing clinical malaria status severity while *E. vermicularis* co-infection showed a declining pattern (Table 1).

### Distribution of cases and controls

There were 143 possible cases and 94 possible controls available for matching as shown in Fig 1.

After running the program, 73 cases and 89 controls were selected in the ratios of 1:1 (57:57) and 1:2 (16:32) as shown in Fig 1. The baseline characteristics of the selected cases and controls including the matching variables are described in Table 2. There were no significant differences in gender, nutritional status, bednets and antihelminth use among cases and controls.

### Effect of soil transmitted helminth on malaria clinical status

In simple conditional logistic regression analysis, there was a tendency for a protective effect of helminth on the development of clinical malaria [OR = 0.6, 95% CI of 0.3–1.3] which was significant for *E. vermicularis* species [OR = 0.2, 95% CI of 0.0–0.9]. On the contrary, there was a tendency of hookworm species to be associated with clinical malaria [OR = 3.0, 95% CI of 0.9–9.5], Table 3.

In multiple conditional regression analysis, the overall protective effect was lower with all helminth infection [OR = 0.8, 95% CI of 0.3–1.9] but remained significantly protective with *E. vermicularis* species [OR = 0.1, 95% CI of 0.0–1.0]. Hookworm species showed was borderline significant with an increased odd towards malaria disease [OR = 3.6, 95% CI of 0.9–14.3], Table 4.

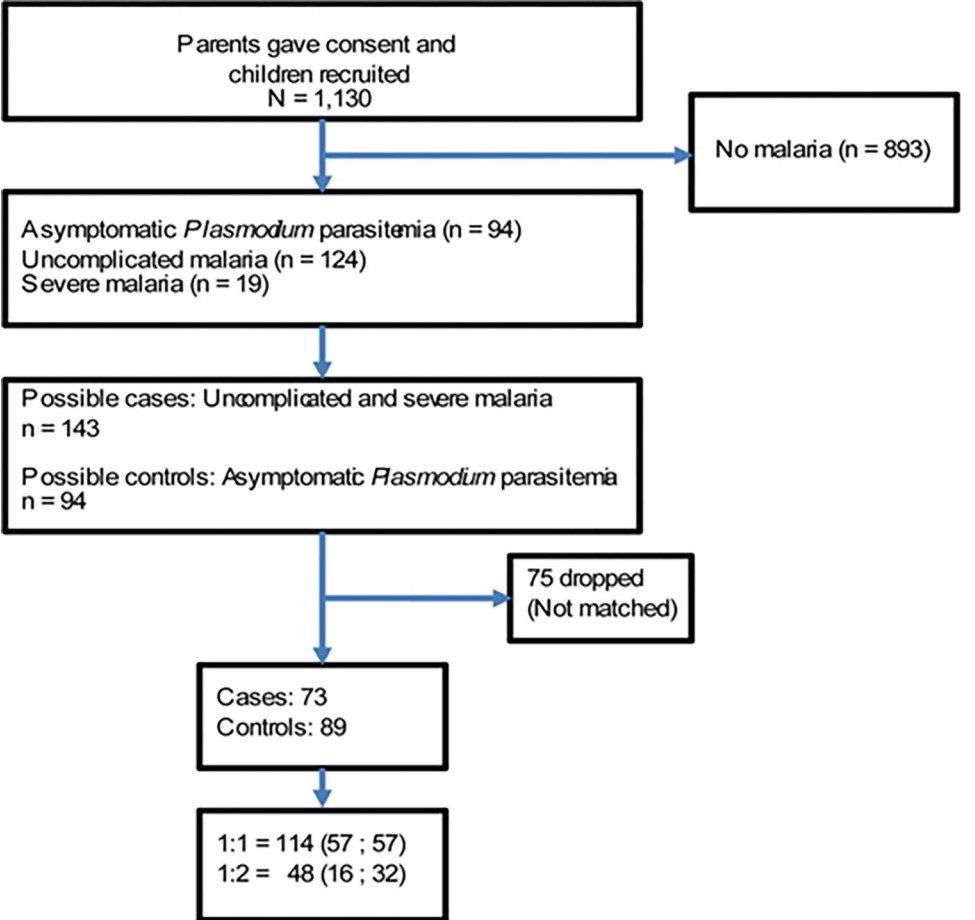

**Fig 1. Flow diagram of the participants and matching procedures.**

Using ordinal logistic regression which better reflects the progression of asymptomatic *Plasmodium* parasitemia to severe malaria, there was a 50% protective effect with overall helminth [OR = 0.5, 95% CI of 0.3–0.9]. On the contrary, hookworm species was highly predictive of clinical malaria [OR = 7.8, 95% CI of 1.8–33.9 for uncomplicated malaria] and [OR = 49.7 (1.9–1298.9) for severe malaria] (Table 5). Fig 2 shows two different model applied to investigate the effect of STH on malaria clinical status from asymptomatic to severe. Fig 2A highlights how the prevalence of helminth infection changes according to malaria clinical status using all possible cases and controls before matching and Fig 2B shows how the prevalence of STH species varies with clinical malaria status in a proportional way among cases and controls using matched data (n = 73 cases and 89 controls). There was a tendency for all STH infections combined to protect against clinical malaria. Dissected for distinct STH infections, this protection became particularly eminent for *E. vermicularis* infection while hookworm infections again enhanced the risk of developing malaria disease (uncomplicated and severe malaria).

## Effect of soil transmitted helminth infection on response to treatment

**Adequate clinical and parasitological response (ACPR).** The crude ACPRs were 62/65 (95.4%) and 156/163 (95.7%) for STH positive and helminth negative children respectively. Most of the recurrent infections were late clinical failure (LCF). Out of a total 10 recurrent

**Table 1. Prevalence of helminth infection among children according to malaria clinical status in Bagamoyo District, Tanzania (before matching).**

| Variables | Total N = 228 | Malaria infection status | | |
| --- | --- | --- | --- | --- |
| | | Asymptomatic n = 92 | Uncomplicated n = 118 | Severe n = 18 |
| All helminth | | | | |
| **Helminth (+ve)** | 65 (28.5) | 34 (37.0) | 27 (22.9) | 4 (22.2) |
| **Helminth (-ve)** | 163 (71.5) | 58 (63.0) | 91 (77.1) | 14 (77.8) |
| Helminth infection | | | | |
| **Single species** | 56 (24.6) | 28 (30.4) | 24 (20.3) | 4 (22.2) |
| **Double species** | 9 (3.9) | 6 (6.5) | 3 (2.5) | 0 (0.0) |
| > **2 species** | 0 (0.0) | 0 (0.0) | 0 (0.0) | 0 (0.0) |
| Single helminth infection | | | | |
| *E. vermicularis* | 20 (8.8) | 13 (14.1) | 7 (5.9) | 0 (0.0) |
| Hookworm * | 21 (9.2) | 6 (6.5) | 12 (10.2) | 3 (16.7) |
| *S. stercoralis* | 13 (5.7) | 7 (7.6) | 5 (4.2) | 1 (5.6) |
| *T. trichiura* | 0 (0.0) | 0 (0.0) | 0 (0.0) | 0 (0.0) |
| Helminth intensity | | | | |
| *E. vermicularis****** | | | | |
| **light** | 10 (4.4) | 7 (7.6) | 3 (2.5) | 0 (0.0) |
| **Moderate** | 5 (2.2) | 2 (2.2) | 3 (2.5) | 0 (0.0) |
| **Heavy** | 5 (2.2) | 4 (4.3) | 1 (0.8) | 0 (0.0) |
| Hookworm*** | | | | |
| **light** | 14 (6.1) | 5 (5.4) | 8 (6.8) | 1 (5.5) |
| **Moderate** | 0 (0.0) | 0 (0.0) | 0 (0.0) | 0 (0.0) |
| **Heavy** | 3 (1.3) | 0 (0.0) | 2 (1.7) | 1 (5.5) |
| *T. trichiura****** | | | | |
| **light** | 0 (0.0) | 0 (0.0) | 0 (0.0) | 0 (0.0) |
| **Moderate** | 0 (0.0) | 0 (0.0) | 0 (0.0) | 0 (0.0) |

Nine children excluded because of no stool collected (2 in asymptomatic *Plasmodium* parasitemia, 6 in uncomplicated and 1 in severe malaria).

*p values was significant.

*** The total number differ from that of total single helminth species as some species were isolated using other methods apart from Kato-Katz and hence couldn't be quantified. NOTE: These are prevalence data during the study period, 2011–2012.

infections, 3 (4.6%) were among the STH positive and 7 (4.3%) among the helminth negative children. All were LCF except for 3 late parasitological failures (LPF) in the helminth negative children.

Generally, children infected with STH had higher geometric mean time to first clearance of parasitemia. The time to first clearance was significantly longer for children who presented with severe malaria and co-infected with *E. vermicularis* and hookworm. The geometric mean time was also longer with *S. stercoralis* co-infection, although not statistically significant (Table 6).

## Occurrence of other diseases than malaria

Respiratory infections [pneumonia and upper respiratory tract infections (URTI)] were among the common disease. Malaria clinical status significantly influenced the occurrence of other diseases (p = 0.038) (Table 7).

**Table 2. Baseline characteristics among cases and controls.**

| | Malaria disease status | |
|---|---|---|
| **Characteristics** | **Cases**<br>**N = 73**<br>**n (%)** | **Controls**<br>**N = 89**<br>**n (%)** |
| Matching variables | | |
| Age group | | |
| **< 3 years** | 15 (20.6) | 15 (16.9) |
| **3–5 years** | 13 (17.8) | 14 (15.7) |
| **> 5 years** | 45 (61.6) | 60 (67.4) |
| Location (village) | | |
| **Kiwangwa** | 71 (97.2) | 86 (96.6) |
| **Msata** | 1 (1.4) | 1 (1.1) |
| **Magomeni** | 1 (1.4) | 2 (2.3) |
| Education level | | |
| **Too young** | 15 (20.6) | 15 (16.8) |
| **Not schooling** | 38 (52.0) | 45 (50.6) |
| **Preprimary** | 11 (15.1) | 11 (12.4) |
| **Primary** | 9 (12.3) | 18 (20.2) |
| Other demographics | | |
| Gender | | |
| **Male** | 39 (53.4) | 45 (50.6) |
| **Female** | 34 (46.6) | 44 (49.4) |
| Nutritional status | | |
| **Normal** | 62 (84.9) | 73 (82.0) |
| **Underweight** | 11 (15.1) | 16 (18.0) |
| **Normal** | 56 (76.7) | 63 (70.8) |
| **Stunted** | 17 (23.3) | 26 (29.2) |
| **Normal** | 73 (100.0) | 86 (96.6) |
| **Wasted** | 0 (0.0) | 3 (3.4) |
| Intervention coverage | | |
| Bednets | | |
| **Slept under a bednet last night** | 61 (88.4) | 74 (88.1) |
| Antihelminth | | |
| **Used albendazole** | 25 (39.1) | 40 (44.9) |
| **Used Mebendazole** | 15 (23.4) | 16 (18.0) |

Malaria disease status: Cases are children who had uncomplicated and severe malaria (disease) and controls are children with asymptomatic *Plasmodium* parasitemia infection.

## Discussion

To our knowledge, this is the first case control study to investigate the effect of soil transmitted helminth (STH), and specifically *E. vermicularis* and *S. stercoralis* on clinical malaria status, with the whole spectrum from asymptomatic *Plasmodium* infection to uncomplicated and severe malaria.

The results of this study showed an overall tendency of a protective effect with any helminth but opposite results with *E. vermicularis* and hookworm species, which may partly explain the contradictory results and controversy in the literature about the association of STH and malaria. Our study shows that co-infection with hookworm is associated with clinical malaria,

**Table 3. Strength of association between malaria disease and helminth infection using simple conditional logistic model.**

| Variables | Cases<br>N = 73<br>n (%) | Controls<br>N = 89<br>n (%) | OR (95% CI) | p—value |
|---|---|---|---|---|
| Helminth# | | | | |
| Helminth (+ve) | 19 (27.5) | 34 (39.1) | 0.6 (0.3–1.3) | 0.241 |
| Helminth (-ve) | 50 (72.5) | 53 (60.9) | | |
| Single helminth species (+ve) | | | | |
| Hookworm | 11 (15.1) | 6 (6.7) | 3.0 (0.9–9.5) | 0.065 |
| *S. stercoralis* | 4 (5.5) | 7 (7.9) | 0.6 (0.1–2.5) | 0.501 |
| *E. vermicularis* | 2 (2.7) | 13 (14.6) | 0.2 (0.0–0.9) | 0.037 |

\# Total number of cases was 69 and controls were 87.

and even more so with severe malaria, while *E. vermicularis* protects against development of clinical and severe malaria. *S. stercoralis* species tends to be protective against clinical malaria although not statistically.

The overall protective effect with any helminth is in line with previous findings as summarized in the systematic reviews [5,42]. In addition, our results are consistent with the findings from previous studies that show hookworm to be a risk factor for uncomplicated and severe malaria [22,24,43–45]. Previous studies have documented the protective effect of *A. lumbricoides* [22,46–49] but none had yet documented such an effect of *E. vermicularis*. It could be that we were able to show a protective effect of this particular species because of the absence of *A. lumbricoides* species in our area, and thanks to the use of sensitive methods to identify *E. vermicularis*.

Immunological analysis of the same children included in this case-control study confirmed the pro-inflammatory effect of hookworm and *S. stercoralis* and the anti-inflammatory effect of *E. vermicularis* [50]. Furthermore, epigenetic analysis showed that *E. vermicularis* acts through the gut microbiota to influence the immune system. Indeed, where the ratio of firmicutes: bacteroidetes is a biomarker of gut inflammation, *E. vermicularis* co-infected children in the same children presented in this epidemiological study had a much higher ratio compared to children co-infected with any helminth and those who had no infection (controls) [50].

*E. vermicularis* is considered a nuisance rather than a cause of serious disease, especially in children [51,52]. Reinfection rate is high and its elimination in the family/community poses significant challenges. The long-term colonization, and non invasive nature of the *E. vermicularis* may limit its aggressive interaction with the immune system creating dormant

**Table 4. Multiple conditional logistic models to assess the effect of helminth on malaria disease.**

| Variables | Malaria disease*<br>OR (95% CI) | p—value |
|---|---|---|
| All helminth | 0.8 (0.3–1.9) | 0.558 |
| Hookworm | 3.6 (0.9–14.3) | 0.063 |
| *S. stercoralis* | 1.0 (0.2–4.9) | 0.984 |
| *E. vermicularis* | 0.1 (0.0–1.0) | 0.049 |

*All models were adjusted for gender, nutritional status, bed net use and use of antihelminth (Albendazole and mendazole).

**Table 5. Strength of association between malaria and helminth infection using ordinal logistic regression model.**

| Explanatory variables | Malaria disease model* | |
|---|---|---|
| | OR (95% CI) | p- value |
| Overall | | |
| **All helminth** | 0.5 (0.3–0.9) | 0.026 |
| **Asymptomatic** | 1 | |
| **Uncomplicated** | 0.5 (0.3–1.0) | 0.065 |
| **Severe** | 0.6 (0.2–2.1) | 0.410 |
| Hookworm | | |
| **Asymptomatic** | 1 | |
| **Uncomplicated** | 7.8 (1.8–33.9) | 0.006 |
| **Severe** | 49.7 (1.9–1298.9) | 0.019 |
| *S. stercoralis* | | |
| **Asymptomatic** | 1 | |
| **Uncomplicated** | 0.3 (0.1–2.1) | 0.247 |
| **Severe** | 0.1 (0.0–2.3) | 0.155 |
| *E. vermicularis* | | |
| **Asymptomatic** | 1 | |
| **Uncomplicated** | 0.6 (0.2–2.0) | 0.381 |
| **Severe** | - | 0.996 |

Model for clinical malaria compares children with clinical malaria (uncomplicated and severe malaria) as cases and those with asymptomatic *Plasmodium* parasitemia infection as controls.

*The model was adjusted for gender, age group, location and education level.

coexistence with its host to survive. The hypothesis that *E. vermicularis* might provide one of the links in the "hygiene hypothesis" by modifying the gut microbiome towards anti-inflammation and many other ways [52] deserves further studies in different human populations.

The different pathogenesis mechanisms of hookworm and *E. vermicularis* could explain the observed opposite effect of the two species. While hookworm is an invasive blood sucking, migratory species associated with negative effect on health [2], *E. vermicularis* is rarely invasive, mostly asymptomatic and symbiotic commensal within the human body [52]. This could indicate the adaptive mechanism in terms of degree of immune stimulation in relation to

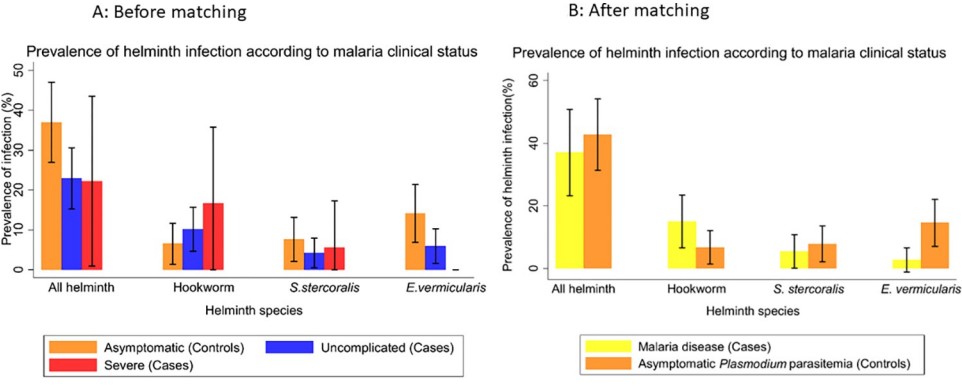

**Fig 2.** Prevalence of helminth infection according to the two models 2A. Malaria clinical status before matching and 2B. Malaria clinical status after matching.

**Table 6. Geometric mean time (in hours) to first clearance of malaria parasitemia according to helminth species.**

| Variables | All helmith** | | E. vermicularis** | | Hookworm** | | S. stercoralis** | |
|---|---|---|---|---|---|---|---|---|
| | Coefficient (95% CI) | βco. | Coefficient (95% CI) | βco. | Coefficient (95% CI) | βco. | Coefficient (95% CI) | βco. |
| Overall | 0.04 (-0.17–0.25) | 1.04 | 0.31 (- 0.17–0.79) | 1.36 | - 0.18 (- 0.61–0.25) | 0.83 | 0.49 (- 0.09–1.07) | 1.63 |
| Asymptomatic (Ref) | | | | | | | | |
| Uncomplicated | 0.04 (- 0.17–0.25) | 1.04 | 0.06 (- 0.29–0.4) | 1.06 | 0.22 (- 0.16–0.60) | 1.24 | 0.06 (- 0.27–0.39) | 1.06 |
| Severe | 1.17* (0.77–1.57) | 3.21 | 1.87* (0.85–2.89) | 6.5 | 1.90* (0.86–2.93) | 6.67 | 1.42 (0.28–2.56) | 4.12 |

βco. denotes beta coefficient (mean time in hours). Note:

**all the four models were adjusted for age group.

* p values were significant

impact of a co-existing species such as *Plasmodium* parasite within the same host. *E. vermicularis* infection is most likely associated with the induction of a T helper 2 (Th2) / regulatory T cells (Treg) hypo-response with an increase release of immunoglobulin E (IgE) complexes that activate high affinity Fc receptors (CD23) and the anti-inflammatory interleukin-10 (IL-10) which activates the nitric oxide synthase releasing nitric oxide leading to reduced sequestration of parasitized red blood cells [53]. On the other hand, hookworm induce the Th2/Treg response which leads to suppression of Th1 and pro-inflammatory responses, which are key for *Plasmodium* parasite clearance within the human body [53]. Additionally, hookworm could alter antibody dependent cellular inhibition (ADCI) with the predominance of non-cytophilic IgG2, IgG4 and IgM while decreasing the cytophilic antibodies IgG1 and IgG3 leading to increased severity of malaria [1,19]. An association between total IgG and hookworm infection in children have been reported in Ghana [44].

The overall protective or enhancing effect of helminths on clinical malaria is obviously driven by the most common STH species in the region studied. We observed that the proportion of hookworm increased as severity of malaria increased, while that of *E. vermicularis* showed the opposite trend. In the present study, the trend of *S. stercoralis* was less clear but its effect was more marked on asymptomatic *Plasmodium* parasitemia as previously reported [31]. The low numbers limited our analysis and hence could hinder to identify the effect of *S. stercoralis* on clinical malaria.

**Table 7. Occurrence of other diseases stratified by helminth status.**

| | Helminth status | | |
|---|---|---|---|
| Diagnosis | Positive | Negative | OR |
| Pneumonia | 4 (18.2) | 21 (35.6) | 0.4 |
| URTI | 7 (31.8) | 19 (32.2) | 1.0 |
| Gastroenteritis | 2 (9.1) | 7 (11.9) | 0.7 |
| Ill-defined illness | 3 (13.6) | 4 (6.8) | 2.2 |
| Others | 6 (27.3) | 8 (13.6) | 2.4 |
| Total | 22 (33.8) | 59 (36.2) | |

Note: URTI means upper respiratory tract infections. p values were not significant

Helminth infected children had a higher geometric mean time to first clearance of *Plasmodium* parasitemia, with almost the same adequate clinical and parasitological response to helminth uninfected children. The findings are in line with those of Degarege et al in Ethiopia [21] which showed that STH do not have significant impact on clearance rate of *Plasmodium* parasite. In our study, severe malaria associated with the co-infection influenced the clearance time. To explore the effect, enough sample size is needed to understand several other factors in a severely sick child.

The utilization of a range of diagnostic tests enabled us to explore a wide range of effect between STH and malaria. There is a need to use adhesive tapes in epidemiological studies in order to detect *E. vermicularis*. Accurate assessment of the respective burden of all STH species is needed to properly assess effect of STH infections on clinical malaria and tailor control programs, in particular for the use of the most appropriate antiparasitic medication that would optimize the public health benefit of deworming and antimalarial programs. One limitation of this study was the low number of severe malaria and the low infection intensity, which reduced the power of our study to investigate morbidity trends. These could be investigated further with the efforts of the ongoing malaria and STH interventions through the National malaria control program (NMCP) and the Neglected tropical disease control program (NTDCP) in Tanzania.

## Conclusion

Overall, these results demonstrate a protective role of *E. vermicularis* against clinical malaria and confirm the enhancing effect of hookworm on malaria morbidity. The protective effect questions of course the progression of deworming programs in relation to annoying consequences of *E. vermicularis* which go against the momentum of elimination. The most affected children live in impoverished societies. The infections cause high morbidity and quantification of the beneficial effect of deworming on economic, school performance, school attendance, cognitive function and overall health status is difficult. Considering the burden of diseases, access and cost, deworming program should thus continue. The present findings should foster the implementation of an integrated control program for these two common parasites coupled with screening and then treating the affected children. The protective effect of *E. vermicularis* highlights the importance of diagnosing this infection and entail further studies to understand better its impact in low and high intensity areas including all ages and at-risk groups to better advice our control programs.

## Supporting information

**S1 Checklist. STROBE checklist.**
(DOC)

**S1 File. Dataset.**
(XLSX)

## Acknowledgments

The authors acknowledge all staff of the IDEA project and all children and their parents who agreed to participate in this study. We are particularly obliged to Raymond Singo, Shabani Halfan, Rehema Mangoli, and Tatu Nassor for their help in the field and laboratory work, the data unit of the BRTC for entering the large amount of data and providing ample support. In addition, we acknowledge the Bagamoyo district officials, specifically the district immunization and vaccination officer (DIVO) Farah Mohammed for his great collaboration.

## Author Contributions

**Conceptualization:** Nahya Salim Masoud, Stefanie Knopp, Salim Abdulla, Marcel Tanner, Claudia Daubenberger, Blaise Genton.

**Data curation:** Nahya Salim Masoud, Stefanie Knopp, Nicole Lenz, Omar Lweno, Ummi Abdul Kibondo, Ali Mohamed, Tobias Schindler, Julian Rothen.

**Formal analysis:** Nahya Salim Masoud, Ummi Abdul Kibondo, Ali Mohamed.

**Funding acquisition:** Salim Abdulla, Marcel Tanner, Claudia Daubenberger, Blaise Genton.

**Investigation:** Nahya Salim Masoud, Stefanie Knopp, Tobias Schindler, Julian Rothen, John Masimba, Alisa S. Mohammed, Fabrice Althaus, Claudia Daubenberger.

**Methodology:** Nahya Salim Masoud, Blaise Genton.

**Project administration:** Nahya Salim Masoud, John Masimba, Alisa S. Mohammed, Claudia Daubenberger.

**Supervision:** Blaise Genton.

**Writing – original draft:** Nahya Salim Masoud.

**Writing – review & editing:** Nahya Salim Masoud, Stefanie Knopp, Nicole Lenz, Omar Lweno, Ummi Abdul Kibondo, Ali Mohamed, Tobias Schindler, Julian Rothen, John Masimba, Alisa S. Mohammed, Fabrice Althaus, Salim Abdulla, Marcel Tanner, Claudia Daubenberger, Blaise Genton.

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
