## [Decision Letter · Decision Letter 0]

1 Jul 2024

Dear Dr. Salim Masoud,

Thank you very much for submitting your manuscript "The impact of soil transmitted helminth on malaria clinical presentation and treatment outcome: A case control study among children in Bagamoyo district, coastal region of Tanzania" for consideration at PLOS Neglected Tropical Diseases. As with all papers reviewed by the journal, your manuscript was reviewed by members of the editorial board and by several independent reviewers. The reviewers appreciated the attention to an important topic. Based on the reviews, we are likely to accept this manuscript for publication, providing that you modify the manuscript according to the review recommendations. 

Sincerely,

David Joseph Diemert, M.D.

Academic Editor

Uriel Koziol

Section Editor

Reviewer's Responses to Questions

**Key Review Criteria Required for Acceptance?**

**Methods**

-Are the objectives of the study clearly articulated with a clear testable hypothesis stated?

-Is the study design appropriate to address the stated objectives?

-Is the population clearly described and appropriate for the hypothesis being tested?

-Is the sample size sufficient to ensure adequate power to address the hypothesis being tested?

-Were correct statistical analysis used to support conclusions?

-Are there concerns about ethical or regulatory requirements being met?

Reviewer #1: The objectives of the study are clearly stated,

The study design is appropriate but there is no power estimation or sample size calculation

The study population is not very clear the authors are either using helminth/soil transmitted helminth infections in the backgound , methods and results and discussion section

The statistical analysis are papropriate but should be discussed in the disccussion section

Reviewer #2: clear objectives

study design clearly stated

**Results**

-Does the analysis presented match the analysis plan?

-Are the results clearly and completely presented?

-Are the figures (Tables, Images) of sufficient quality for clarity?

Reviewer #1: The analysis well presented with monor issues for example Table 1 some variables like T.trichiura are included while they were not observed in the study i.e with zero values

Here there is a mix of helminth and STH, Is E. vemicularis an STH?

Figures are clear

Reviewer #2: clear and complete results

**Conclusions**

-Are the conclusions supported by the data presented?

-Are the limitations of analysis clearly described?

-Do the authors discuss how these data can be helpful to advance our understanding of the topic under study?

-Is public health relevance addressed?

Reviewer #1: The conclusion are based on the findings , limitations not stated , also choice of different statisticla regression methods should be discussed.

Why there was no A.lumbricoides in the study sites,should be discussed

In line 459 Discussion is confusing

"The findings are in line with those of Degarege et al in Ethiopia [16] which showed that STH do not have significant impact on clearance rate of Plasmodium parasite. In ourstudy, severe malaria associated with the co-infection i 460 nfluenced the clearance time." please clarify

Reviewer #2: no limitation of the study stated

**Editorial and Data Presentation Modifications?**

Reviewer #1: The background section should be revised the way it is organised it is difficult to follow some citations with no topical sentences makes difficult to understand one example 'Northern shore of lake Victoria where malaria is endemic showed an increased burden of childhood malaria morbidy associated with hookworm helminth infection during pregnancy'

they should start with literature which support their hypothesis followed negative and then mixed view points

Reviewer #2: no suggestions

**Summary and General Comments**

Reviewer #1: (No Response)

Reviewer #2: The article is well designed scientifically and methodologically, my only concern is why waiting until this year to publish the article since the study was done in 2011/2012?

PLOS authors have the option to publish the peer review history of their article (what does this mean?). If published, this will include your full peer review and any attached files.

Reviewer #1: No

Reviewer #2: No

Figure Files:

Data Requirements:

Reproducibility:

References

---

## [Decision Letter · Decision Letter 1]

29 Jul 2024

Dear Dr. Salim Masoud,

We are pleased to inform you that your manuscript 'The impact of soil transmitted helminth on malaria clinical presentation and treatment outcome: A case control study among children in Bagamoyo district, coastal region of Tanzania' has been provisionally accepted for publication in PLOS Neglected Tropical Diseases.

Best regards,

David Joseph Diemert, M.D.

Academic Editor

Uriel Koziol

Section Editor

Reviewer's Responses to Questions

**Key Review Criteria Required for Acceptance?**

**Methods**

-Are the objectives of the study clearly articulated with a clear testable hypothesis stated?

-Is the study design appropriate to address the stated objectives?

-Is the population clearly described and appropriate for the hypothesis being tested?

-Is the sample size sufficient to ensure adequate power to address the hypothesis being tested?

-Were correct statistical analysis used to support conclusions?

-Are there concerns about ethical or regulatory requirements being met?

Reviewer #1: The authors have addressed all questions in this section

**Results**

-Does the analysis presented match the analysis plan?

-Are the results clearly and completely presented?

-Are the figures (Tables, Images) of sufficient quality for clarity?

Reviewer #1: The analysis is clear and results clearly presented

**Conclusions**

-Are the conclusions supported by the data presented?

-Are the limitations of analysis clearly described?

-Do the authors discuss how these data can be helpful to advance our understanding of the topic under study?

-Is public health relevance addressed?

Reviewer #1: The conclusion support the study objectives and findings of the study

**Editorial and Data Presentation Modifications?**

Reviewer #1: None

**Summary and General Comments**

Reviewer #1: The paper is timely and provides more evidence on the diagnosis and treatment of helminth infections

PLOS authors have the option to publish the peer review history of their article (what does this mean?). If published, this will include your full peer review and any attached files.

Reviewer #1: **Yes: **Billy Ephraim Ngasala

---

## [Editor Report · Acceptance letter]

6 Aug 2024

Dear Dr. Salim Masoud,

We are delighted to inform you that your manuscript, "The impact of soil transmitted helminth on malaria clinical presentation and treatment outcome: A case control study among children in Bagamoyo district, coastal region of Tanzania," has been formally accepted for publication in PLOS Neglected Tropical Diseases.

Best regards,

Shaden Kamhawi

co-Editor-in-Chief

Paul Brindley

co-Editor-in-Chief
